# Antioxidant Activity of Different Tissues from Rabbits Fed Dietary Bovine Colostrum Supplementation

**DOI:** 10.3390/ani13050850

**Published:** 2023-02-26

**Authors:** Valentina Serra, Marta Castrica, Stella Agradi, Giulio Curone, Daniele Vigo, Alessia Di Giancamillo, Silvia Clotilde Modina, Federica Riva, Claudia Maria Balzaretti, Roberta De Bellis, Gabriele Brecchia, Grazia Pastorelli

**Affiliations:** 1Department of Veterinary Medicine and Animal Sciences, University of Milan, Via dell’Università 6, 26900 Lodi, Italy; 2Department of Biomedical Sciences for Health, University of Milan, Via Mangiagalli 31, 20133 Milan, Italy; 3Department of Biomolecular Sciences, University of Urbino “Carlo Bo”, Via A. Saffi 2, 61029 Urbino, Italy

**Keywords:** rabbit, antioxidant status, bovine colostrum, gene expression

## Abstract

**Simple Summary:**

Bovine colostrum (BC) is the first secretion of mammary glands produced after birth; it represents a natural source of nutrients essential for the growth and development of newborns. Given its various properties, including antioxidants, this study intended to determine the effects of the dietary supplementation of two different concentrations (2.5% and 5%) of BC on the antioxidant status in different tissues (blood, liver and muscle) of rabbits. No differences in dietary treatment were recorded regarding the plasma concentration of antioxidant enzymes (catalase, glutathione peroxidase and superoxide dismutase), or regarding the gene expression of the aforementioned enzymes in the liver and longissimus dorsi (LD) muscle samples of rabbits. Further studies are needed to better understand the effect of this potentially promising nutraceutical in rabbit meat.

**Abstract:**

Recent advances in animal nutrition have indicated that bovine colostrum (BC), due to its content of macronutrients, micronutrients and bioactive compounds, is an excellent health supplement. To the best of our knowledge, no studies on the effect of BC on antioxidant status have been performed in rabbits. This study aimed to investigate the effect of two BC concentrations on antioxidant status and gene expression of antioxidant enzymes in some tissues of rabbits. Thirty New Zealand White male rabbits were randomly divided into three experimental diets, containing 0% (CON), 2.5%, and 5% of BC (BC-2.5 and BC-5, respectively). The activity of antioxidant enzymes in plasma (catalase: CAT; glutathione peroxidase: GPx; superoxide dismutase: SOD), and the enzymes’ gene expression in the liver and longissimus dorsi muscle, were determined. Results showed no significant differences, neither in plasma nor in tissues. A significant tissue-related effect has been observed regarding the mRNA levels of SOD and GPx, which were higher in the LD (*p* = 0.022) and liver (*p* = 0.001), respectively. Further studies, considering modifications of the length and dosage of dietary BC supplementation, are required to update the current state of knowledge in rabbits, as well as to fully understand the potential value of BC for possible application in farming use.

## 1. Introduction

Colostrum, also known as first milk, is produced by the mammary gland immediately following parturition. Bovine colostrum (BC), due to its role in immunity, growth and antimicrobial factors, has been used for centuries as a traditional or complementary therapy for a wide variety of disorders and in veterinary practice. Its role for a newborn is not only to provide nutrition for the first few days of life, but also to provide protection against infections while the immune system is still developing [1]. Several studies support its beneficial activity for the treatment of medical conditions in children and adults, particularly gastrointestinal disorders [2,3]. Its use is not limited to humans; in fact, its role as an effective nutraceutical for the enhancement of the immune function in several animal species, both farm and pet, has been documented [4,5,6,7]. Colostrum is also rich in enzymatic (lactoperoxidase, catalase, superoxide dismutase, glutathione peroxidase) as well as non-enzymatic (vitamins E, A, C, lactoferrin and selenium) antioxidants, which confer it antioxidant activity. The evaluation of the potential antioxidant effects of BC dietary supplementation in rabbit species could be relevant, due to the high proportion of poly-unsaturated fatty acids (PUFAs) that rabbit meat contains [8].

For the rabbit meat industry, Italy represents the third-largest producer within the European Union, after Spain and France. Although Italy is among the top five producers of rabbit meat in the world, its consumption has decreased over the past five years in the Mediterranean region [9]. Rabbits are sensitive to many bacterial infections, such as respiratory and intestinal diseases, which cause **a** high mortality of the animals, thus weighing heavily on the productivity of farms [10]. These pathologies have multifactorial etiologies, connected to diet, hygiene, environmental and climatic factors, and stress. Oxidative stress occurs when the production of potentially destructive reactive oxygen species (ROS) exceeds the body’s natural antioxidant defense [11]. Oxidative stress can be measured directly, by detecting free radical production, or indirectly, by detecting the antioxidant defenses of the organism. Since oxidative stress is involved in several pathological conditions in farm animals, the evaluation of oxidative status is a good indicator of the health status of animals. Antioxidant enzyme activities are generally used for the evaluation of oxidative stress status in animals [12]. Superoxide dismutase (SOD), catalase (CAT) and glutathione peroxidase (GPx) enzymes represent the primary components of the enzymatic antioxidant defense system against oxidative stress.

Dietary antioxidant supplementation is considered a promising way for mitigate oxidative stress. For this reason, the additional supply of antioxidant substances is able to contribute to the direct and indirect extinction of free radicals, complementing the activities of primary antioxidant enzymes [13]. Currently, great attention is paid to natural products with antioxidant effects, of which BC is a valuable example. To the best of our knowledge, no information about the effects of BC on blood antioxidant status and antioxidant gene expression in rabbits is available. Based on the positive antioxidant activity of BC described in the literature in piglets and calves [14,15], we hypothesized that a similar effect could also be exerted in rabbits.

The study was planned to evaluate the effect of two increasing levels of dietary BC on plasma antioxidant enzymes’ activity and the expression of primary antioxidant enzymes in the liver and longissimus dorsi muscle of rabbits.

## 2. Materials and Methods

### 2.1. Animals Diets and Samples Collection

The experimental protocol was conducted at the Department of Agricultural, Food, and Environmental Science of the University of Perugia’s experimental farm, in compliance with the guidelines of the Legislative Decree No. 146, implementing Directive 98/58/EC on the protection of animals bred for farming purposes.

A total of 30 post-weaning New Zealand White male rabbits (37 days old), with an average body weight of 550 ± 24 g, were randomly assigned to three experimental diets: a control group (CON) fed with a pelletted commercial standard diet, CON supplemented with 2.5% of liquid bovine colostrum (BC-2.5), and CON supplemented with 5.0% of liquid bovine colostrum (BC-5) (Table 1). Rabbits were fed ad libitum and had free access to water. The experimental colostrum, of high quality (>23% Brix; [16]), was collected from multiparous Holstein–Friesian cows during the first milking, and immediately stored in sterile containers at −20°C until the start of the experiment. All rabbits were kept under the same management, and the same hygienic and environmental conditions; their clinical condition was monitered throughout the trial. Thirty-seven days after the beginning of the feeding trial, the animals were slaughtered (body weight of 1977 ± 26 g), and fasting blood samples were collected by jugular vein puncture using vacutainers. The blood was then centrifuged at 3000× *g* at 4 °C for 15 min to separate the plasma; the plasma aliquots were stored at −80 °C until subsequent analyses.

The carcasses were left for 24 h at 2.5 ± 0.5 °C, and the LD muscle (left side) and liver (a portion of the right lobe) were taken from each animal, for a total of 60 samples. Samples were stored in RNAlater™ (ThermoFisher Scientific, Waltham, MA, USA) soon after the sampling procedure, and maintained at -80°C until their subsequent use in RNA extraction.

### 2.2. Total Phenol Content and Antioxidant Activity of Bovine Colostrum and Experimental Diets

#### 2.2.1. Sample Treatment

All samples were extracted using 70% (v/v) ethanol/double-distilled water. The BC sample was extracted at a ratio of 1 mL in 2 mL 70% EtOH, while finely ground samples of control diet, and control diet supplemented with 2.5% and 5% of BC, were extracted at a ratio of 2 g in 10 mL of 70% EtOH (CON, BC-2.5, and BC-5, respectively). All samples were shaken for 1 h in the dark, and were centrifuged at 13,000 rpm for 15 min at 4 °C. Recovered supernatants were stored at −20 °C until analyses.

#### 2.2.2. Total Phenol Content

The Folin-Ciocalteu method was used to determine total phenol contents (TPs) [17]. Different concentrations of gallic acid (from 2 to 10 μg/mL) were used as a standard to construct the calibration curve, and values were expressed as milligrams of gallic acid equivalents: (GAE)/mL for BC, and GAE/g for CON, BC-2.5, and BC-5 samples.

#### 2.2.3. Antioxidant Activities (DPPH, ABTS, ORAC Assays)

The antioxidative properties of BC and experimental diets were measured in terms of the Trolox equivalent antioxidant capacity (TEAC), using DPPH, ABTS and ORAC assays. The DPPH assay was carried out as reported in Saltarelli et al. [18], with minor modifications. Briefly, 850 μL of 100 μM 2,2-diphenyl-1-picrylhydrazyl (DPPH) ethanol solution was added to 150 μL of sample at various concentrations. Samples were rested for 20 min in the dark at room temperature, and then the absorbance, decreasing at 517 nm, was detected (UV Beckman Spectrophotometer CA, USA); 70% (v/v) ethanol/double-distilled water was used as a blank. The DPPH radical-scavenging activity (DSA) of the sample was calculated as follows: DSA (%) = [(blank Abs − sample Abs)/ blank Abs] × 100.

For the ABTS assay [19], 10 μL of the samples and 1 mL of 2,2,-azino-bis(3-ethylbenzothiazoline-6-sulphonic acid) ethanolic solution were left for 5 min in the dark at room temperature, and then the absorbance was detected at 734 nm (UV Beckman spectrophotometer CA, USA); 70% (v/v) ethanol/double-distilled water was used as blank. The ABTS radical-scavenging activity (ASA) was reported as follows: ASA (%) = [(blank Abs − sample Abs)/ blank Abs] × 100.

The oxygen radical absorbance capacity (ORAC) was performed as in De Bellis et al. [20], detecting fluorescence until total extinction (485 nm ex. and 520 nm em.) on a Fluostar Optima plate reader (BMG Labtech, Offenburg, Germany).

All antioxidant assays were compared to the Trolox antioxidant capacity, and, for this reason, data are reported as the micromoles of Trolox equivalents (TE)/mL, or TE/mg, of the samples.

### 2.3. Antioxidant Enzymes Activity Determination

The activity of catalase (CAT), glutathione peroxide (GPx) and superoxide dismutase (SOD) enzymes was tested in plasma samples using commercial kits supplied by Cayman Chemical (Ann Arbor, MI, USA), according to the instructions provided by the manufacturer. CAT activity in plasma was measured with the Catalase Assay Kit (ref. No.: 707002, Cayman Chemicals, Ann Arbor, MI, USA), which utilizes the peroxidase function of CAT for the determination of enzyme activity. In the presence of an optimal H_2_O_2_ concentration, the CAT enzyme reacts with methanol, producing formaldehyde, which is measured colorimetrically with the chromogen 4-amino-3-hydrazino-5-mercapto-1,2,4-triazole (Purpald). This chromogen specifically forms a bicyclic heterocycle with aldehydes, which changes from colorless to purple upon oxidation. One unit is defined as the amount of CAT enzyme that forms 1.0 nmol of formaldehyde per minute at 25°C (according to the manufacturer’s protocol).

GPx activity in plasma was measured with a Glutathione Peroxide Assay Kit (ref. No.: 703102, Cayman Chemicals, Ann Arbor, MI, USA), which measures GPx activity indirectly by a coupled reaction with glutathione reductase. The reduction of hydroperoxide by GPx produces oxidized glutathione, which is recycled to its reduced state by glutathione reductase and NADPH. The latter is oxidized to NADP+, decreasing the absorbance at 340 nm. One unit of GPx is defined as the amount of enzyme that oxidizes 1.0 nmol of NADPH to NADP+ per 1 min at 25 °C (according to ref. No.: 703102, Cayman Chemicals, USA Kit).

The assay of SOD activity was performed using the commercial kit “Superoxide Dismutase Assay Kit” (ref. No.: 706002, Cayman Chemical, Ann Arbor, MI, USA), which utilizes a tetrazolium salt for the detection of superoxide radicals generated by xanthine oxidase and hypoxanthine. The amount of SOD enzyme required to exhibit a 50% dismutation of the superoxide radical represents one unit of SOD (according to the manufacturer’s protocol).

### 2.4. RNA Extraction and cDNA Synthesis

Total RNA was isolated from liver and LD samples using a commercial kit (ReliaPrep™ RNA Tissue Miniprep System; Promega Corporation, USA), following the manufacturer instructions, and was eluted in 30 μL of RNase-free water. The purity and concentration of each RNA sample was measured using a NanoDrop ND-1000 spectrophotometer (ThermoFisher Scientific, Massachusetts, USA). One microgram of each RNA sample was treated with DNase enzyme to remove any genomic DNA contamination using DNase I, RNase-free kit (ThermoFisher Scientific, Waltham, MA, USA). Synthesis of cDNA was performed from 500 ng of total RNA using the GoScript™ Reverse Transcription Mix (Promega Corporation, Madison, WI, USA). For each sample, a no-reverse transcriptase (no-RT) control was used to verify the complete DNA removal. The cDNA samples were stored at −20°C until quantitative real-time PCR analyses.

### 2.5. Real Time RT-qPCR

The gene expression of specific target genes was determined: catalase (CAT), glutathione peroxidase (GPx) and superoxide dismutase (SOD); glyceraldehyde 3-phosphate dehydrogenase (GAPDH) and beta-actin (ACTB) were used as reference genes for real-time quantitative PCR, which was used to compare the expression levels of the three experimental groups (CON, BC-2.5 and BC-5). The primers were designed using the tool Primer-BLAST (https://www.ncbi.nlm.nih.gov/tools/primer-blast/), and were supplied by Eurofins Genomics (Eurofins Genomics, Ebersberg, Germany). Table 2 reports the primer sequences, annealing temperatures (Tm) and amplicon size of each fragment.

cDNA samples were used as templates for the gene expression analysis, which was performed using the thermocycler BioRad iQ5 Real-Time PCR (Bio-Rad, CA, USA), and Universal SYBR^®^ Green Supermix (Bio-Rad, CA, USA) was used as a fluorescent molecule. Each reaction was performed as follows: 500 ng of cDNA, 10 μL of Universal SYBR^®^ Green Supermix, 150 nM of each forward and reverse primer pair (GAPDH, ACTB, and GPx), 250 nM of CAT and SOD genes, and nuclease-free water to reach a final volume of 20 μL. The thermal profiles were as follows: 95 °C for 3 min, 40 cycles of 95 °C for 15 sec, 59 °C (GAPDH, ACTB, GPx, CAT) or 58 °C (SOD) for 30 sec; a melting profile was included after the last amplification cycle to exclude the presence of non-specific amplifications. All data were normalized to the reference genes GAPDH and ACTB, and the relative gene expression of each target gene was calculated using the 2^−ΔΔCt^ method.

### 2.6. Statistical Analyses

Statistical analyses were performed using SPSS 26.0 (SPSS Inc., Chicago, IL, USA). Values are expressed as means ± SD. The distribution of all data was firstly explored for their agreement with normal distribution using the Kolmogorov–Smirnov goodness-of fit test. Where data were not normally distributed, they were log-transformed, and retested for normality before analysis. The effect of treatment and tissue was analyzed by ANOVA. For all analyses, differences were considered significant at *p* < 0.05.

## 3. Results

### 3.1. Total Phenol Content and Antioxidant Activity of Bovine Colostrum and Experimental Diets

The total phenol content (TPC) of BC was equal to 0.084 ± 0.005 mg GAE/mL, while DPPH, ABTS and ORAC assays gave results equal to 0.34 ± 0.05, 2.00 ± 0.13 and 3.92 ± 0.19 μmol/TE, respectively. Table 3 reports the TPC and antioxidant activities found in the three experimental diets (CON, BC-2.5 and BC-5).

### 3.2. Antioxidant Enzymes Activity in Plasma

Antioxidant enzyme activity data are listed in Table 4. No significant differences in plasma CAT, GPx and SOD activities were found among dietary treatments (*p* > 0.05) (Table 4).

### 3.3. Antioxidant Enzymes Gene Expression

The results of relative mRNA expression levels in liver and LD muscle are presented in Table 5. No significant differences were observed in terms of mRNA expression between CAT, GPx and SOD in the liver and LD muscle of the rabbits among the three groups (Figure 1).

A significant tissue-related effect has been observed in terms of the mRNA levels. Indeed, mRNA levels were significantly higher in the LD than the liver for SOD (*p* = 0.022) and GPx (*p* = 0.001) enzymes (Figure 2).

## 4. Discussion

Bovine colostrum (BC) has been widely studied for its antioxidant activity. It is rich in essential nutrients, such as proteins, fats, vitamins and minerals. Additionally, it contains high levels of bioactive compounds, including oligosaccharides, immunoglobulins, lactoferrin, and lysozyme [21], which contribute to passive immunity, antimicrobial protection, and to the development of the gastrointestinal system in the early life of calves [22,23]. It is recognized to provide beneficial nutrients essential for postnatal growth [24].

Due to the lack of knowledge about the potential of this product, the amount of BC produced in excess in dairy farms is usually discarded. Nowadays, researchers have investigated BC’s potential as a product with health benefits, although the utilization of its potential needs to be more effective among the producers.

Firstly, BC for human use has been utilized as a functional ingredient in several food categories, such as cheeses, yoghurts, ice cream, and beverages. There is also evidence that the biologically active proteins present in BC exert not only antibacterial and antiviral activity, but also improve peristalsis and regulate the work of the digestive tract [25]. The positive effects of BC on the human body have also been described during antibiotic therapy; its administration at the beginning of therapy reduced diarrheal episodes and increased bacterial susceptibility to the antibiotic, thus reducing the drug dose.

The use of BC is not confined to humans, but its supplementation as a nutraceutical for both production and companion animals of all ages has been documented. Concerning farm animals, the supplementation of BC in post-weaning piglets improved the GPx activity by 19% and decreased the production of MDA in blood, thus providing supplementary protection against oxidative stress [24].

The dietary supplementation with 0.1% of BC for 40 weeks improved the function of the gut-associated lymphatic tissue of dogs, as higher fecal IgA levels and increased plasmatic responses against canine distemper virus vaccination were observed [6].

The higher antioxidant capacity of colostrum than mature milk depends on presence of a higher level of anti-stress vitamins, such as vitamins A (retinol), E (tocopherol), and C, which are present at higher levels in colostrum than in normal milk [26].

In the current study, antioxidant activities were evaluated by three different methods to perform a comparative overview. The most interesting results came from the ORAC assay, evidencing that BC inclusion in the supplement diet provides better antioxidant activity compared to CON samples.

The highest values of Trolox equivalent indicate the sample’s good ability to protect against the formation of peroxyl radicals, and, therefore, indicate a potentially high antioxidant activity. The antioxidant capacity of BC found herein agrees with the data present in the literature on bovine milk [27], being slightly higher than that reported in human milk [28]; the potential antioxidant activity of diets, as measured by ORAC, increases in the feed according to BC inclusion, confirming the appropriate incorporation of the BC inclusion in the complete diet.

Antioxidants, which are divided in two classes, protect organisms from oxidative damage by free radicals [29]. The enzymes superoxide dismutase (SOD), catalase (CAT) and glutathione peroxidase (GPx) belong to the first category, while the second one is found in exogenous sources, such as dietary supplements [29]. Enzymes such as SOD, CAT and GPx convert free radicals into safer compounds; briefly, SOD reduces superoxide to hydrogen peroxide, and CAT and GPx reduce hydrogen peroxide, via the superoxide dismutase reaction, to water.

Bovine colostrum has been reported to improve serum SOD and GPx activities, while depressing MDA production in piglets [14]. Likewise, the supplementation of BC to Holstein calves has determined an improvement in the antioxidant status, by increasing the SOD activity [15].

In the present study, contrary to our expectations, we found that BC dietary supplementation had no effect on the plasma levels of CAT, GPx and SOD in the rabbits. Nevertheless, a significant tissue-related effect in mRNA levels of SOD and GPx has been found. To the best of our knowledge, the mechanisms by which BC could interact with the antioxidant enzymes are not clear.

Up to now, the lack of literature data on antioxidant enzyme activity in rabbits supplemented with BC makes comparisons and considerations difficult.

A numerical increase of GPx concentrations in the plasma of BC-2.5 group in comparison with the control animals has been observed; the result could be due to an initial activation of the antioxidant response. The activity of the two other main enzymes involved in the antioxidant defense mechanism, SOD and CAT, remained unchanged. We speculate that the weak activity of antioxidant enzymes is due to a sparing effect of dietary antioxidants. Alía et al. [30] reported that, in presence of high concentrations of exogenous antioxidants in the circulatory system, a low demand for enzymatic antioxidant function is required.

Mokhtarzadeh et al. [31] found a significant increase (*p* < 0.01) in SOD serum concentrations in Japanese quails supplemented with BC (2% and 4%) for six weeks.

In our study, dietary supplementation of BC failed to affect CAT, GPx and SOD gene expression levels in the liver and LD muscle of treated rabbits. We speculate that the similar content of phenolic compounds found in all the three experimental diets somehow affected the response of gene expression; the increase in enzyme activity and gene expression found in sheep [32] likely resulted from a combined effect of the n-3 PUFA and the phenolic compounds in the administered perilla seeds. In the present study, all three diets showed similar polyphenol content, according to the responses of the relevant enzymes.

In the organism, dietary compounds must pass several barriers during their metabolic path before being adsorbed. Furthermore, the various biotransformation processes they undergo reduce their bioavailability and accumulation in tissues [33]. In rabbits, the cecum is the site of microbial fermentation, which permits the release of nutrients from ingested food, and the fermentation products are absorbed directly through the wall of the cecum or re-ingested as cecotrophs. The efficiency of the rabbit’s digestion depends, primarily, on the production and ingestion of cecotrophes [34]. Another possible explanation for the lack of alteration in antioxidant enzymes’ gene expression in tissues could be due to the biotransformation processes that antioxidant compounds present in BC undergo, limiting their bioavailability.

Under normal physiological conditions, the oxidation and antioxidant defense systems are in dynamic equilibrium. Our understanding of how cells maintain the capacity to respond to shifts in antioxidant challenges associated with shifting metabolic demands of healthy animals is incomplete, and remains a crucial issue of growth biology.

However, the difference between chronic and oxidative stress has been investigated well [35], and it is known that oxidative stress promotes changes in metabolic balance and the efficiency of nutrient use [36]. In the present study, no oxidative stress was induced.

A hypothesis associated with growing animals is that an increased redox state (generation of more reactive oxygen species than can be reduced by enzymatic and non-enzymatic antioxidant systems) may result in sub-optimal growth [37,38,39,40].

Growth performances of all three groups were in accordance with those of rabbits in trading conditions, with no significant effect (*p* > 0.05) of the dietary treatment on the final body weight (2.06 vs. 1.93 vs. 1.94 kg in CON, BC-2.5 and BC-5 respectively); moreover, in healthy animals, reactive oxygen species’ production is counterbalanced by antioxidant defenses, and an imbalance between their generation and inactivation leads to oxidative stress [41,42]. We speculated that no activation of antioxidant system enzyme has been found, since animals were not subjected to any oxidative stress challenges, and the rabbits remained healthy (absence of clinical signs, absence of lesions, shiny coat) throughout the study.

Important variations in enzyme activity between organs or tissues are usual. For example, variations among SOD and/or CAT levels have been reported in rats by comparing brain, heart, kidney, liver, lung and spleen samples [43], in primates by comparing liver, brain and heart samples [44], and also in rabbit, comparing the liver, brain and testis [45].

In the present work a significant tissue-related effect in mRNA level of SOD and GPx, has been found. The GPx was widely expressed in both tissues examined, but the highest expression was found in the liver, according to the fact that this organ is known to be a rich source of GPx [46].

This antioxidant enzyme normally decreases and detoxifies the hydrogen peroxide and organic hydroperoxides; its mRNA expression is particularly sensitive to any changes in ROS accumulation [47].

GPx activity being higher in the liver than the muscle agrees with a study conducted on rats by Zhang et al. [48], who found the following order of tissue expression: liver > kidney > heart > lung > brain = muscle.

As a potential explanation, we speculate that, due to different oxygen consumption rates, different tissues express distinct antioxidant enzyme activities when subjected to the same treatment [49].

## 5. Conclusions

This study was set out to determine whether increased levels of BC would alter the antioxidant response in rabbits. The results provided information for the first time, on the effect of BC on the antioxidant status of different tissues in rabbits.

The antioxidant activity in BC seems to be confirmed. Anyway, the dietary inclusion of BC into diet produced an unchanged antioxidant defense system in rabbits. Based on this study’s finding, the use of BC at concentrations of 2.5% and 5% did not influence the antioxidant status of the rabbits, since muscular and hepatic expression of the antioxidant enzymes did undergo significant changes. No detrimental or positive alterations have been also observed in terms of the concentrations of antioxidant enzymes in plasma.

Certainly, a study considering some modifications of BC use could be performed, in order to better understand the effect of this potentially promising nutraceutical. Increasing doses, length of supplementation or addition with other factors that act synergistically with BC could be investigated.

## Figures and Tables

**Figure 1 animals-13-00850-f001:**
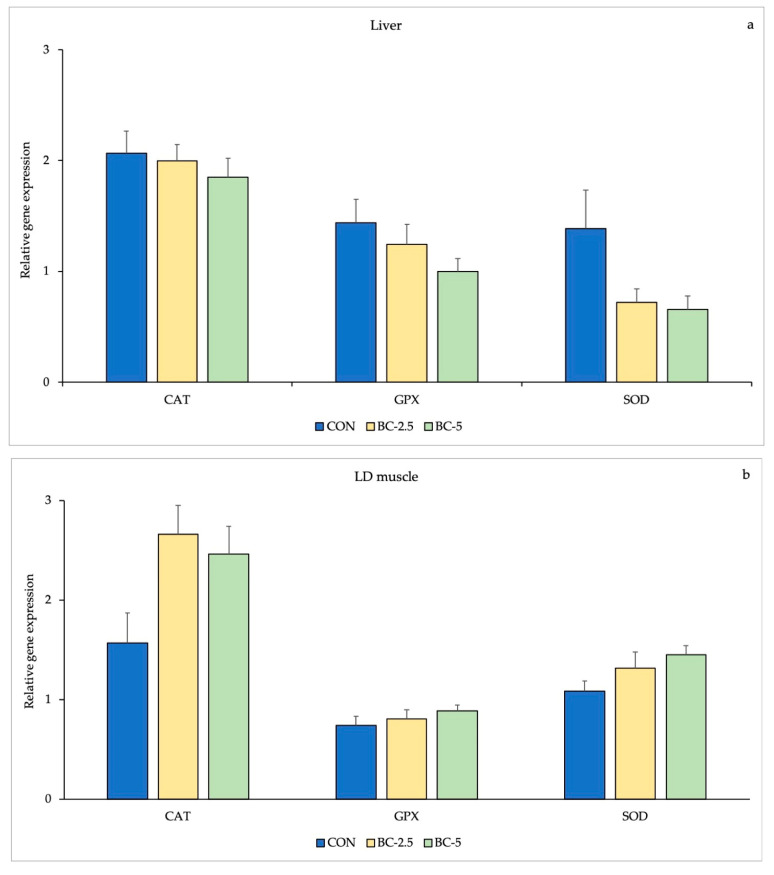
mRNA expression levels of antioxidant enzymes CAT, GPx and SOD in the liver (**a**) and LD muscle (**b**) of rabbits fed diets supplemented with bovine colostrum. The data are reported as means ± SD. CON: group without supplemented diet; BC-2.5: group supplemented with 2.5% of bovine colostrum; BC-5: group supplemented with 5.0% of bovine colostrum; CAT: catalase; GPx: glutathione peroxidase; SOD: superoxide dismutase; SD: standard deviation.

**Figure 2 animals-13-00850-f002:**
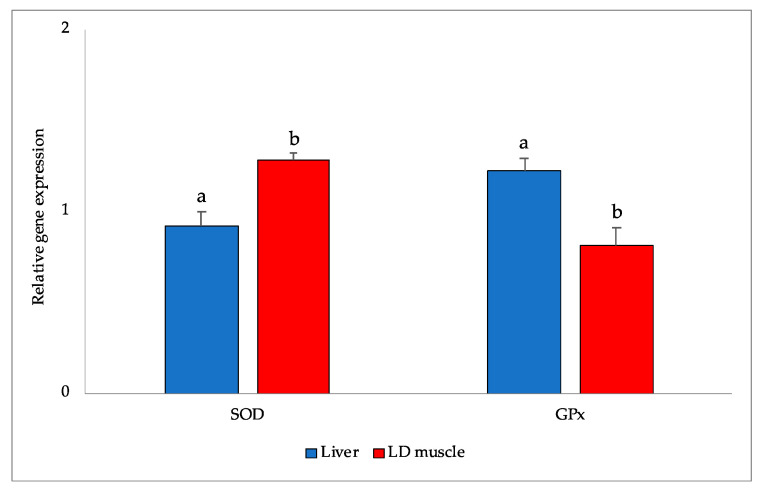
mRNA expression level of SOD and GPx in the liver and LD muscle of rabbits. GPx: glutathione peroxidase; SOD: superoxide dismutase; ^a,b^ Means within a row with different letters are significantly different at *p* < 0.05.

**Table 1 animals-13-00850-t001:** Ingredients (percentage wet weight) of control (CON) and experimental diets of post-weaning rabbits.

Ingredients	Basal Diet	Experimental Diets	
	CON	BC-2.5	BC-5
Dehydrated alfalfa meal	32.40	32.40	32.40
Barley	22.40	19.90	17.40
Wheat bran	20.50	20.50	20.50
Sunflower meal	8.00	8.00	8.00
Soybean meal	4.00	4.00	4.00
Cane molasses	3.00	3.00	3.00
Carob pods	2.70	2.70	2.70
Wheat meal	2.50	2.50	2.50
Calcium Carbonate	2.00	2.00	2.00
Vitamin–mineral premix ^1^	1.60	1.60	1.60
Soybean oil	0.50	0.50	0.50
Sodium chloride	0.40	0.40	0.40
Bovine colostrum	-	2.50	5.00
Chemical composition ^2^			
Dry matter	92.34	91.71	91.69
Crude protein	14.82	14.76	15.23
Ether extract	2.79	2.95	3.02
Ash	7.04	7.23	7.62
NDF	40.00	36.81	35.79
ADF	27.04	24.92	24.31
ADL	12.02	10.03	9.11

CON: group without supplemented diet; BC-2.5: CON supplemented with 2.5% of bovine colostrum; BC-5: CON supplemented with 5.0% of bovine colostrum; NDF: neutral detergent fiber; ADF: acid detergent fiber; ADL: acid detergent lignin. ^1^ Contains, per kg of feed: retinyl acetate, 4800 IU; vitamin D3, 1800 IU; calcium-D pantothenate, 6.6 mg; choline chloride, 300 mg; ferrous sulfate monohydrate, 37.5 mg; manganous sulphate monohydrate, 54.0 mg; copper sulphate pentahydrate, 6.0 mg; zinc oxide, 36.0 mg; potassium iodide, 0.66 mg; sodium selenite, 0.12 mg; butylhydroxytoluene, 30.0 mg; butylhydroxyanisole, 30.0 mg.^2^ Analyzed.

**Table 2 animals-13-00850-t002:** Primers used for quantitative real-time PCR analysis of gene expression in the liver and *L. dorsi* muscle of this study.

Gene	Forward Primer	Reverse Primer	Tm F (°C)	Tm R (°C)	Amplicon Size (bp)
GAPDH	GTCAAGGCTGAGAACGGGAA	TCTCCATGGTGGTGAAGACG	59.4	59.4	142
ACTB	ACATGGAGAAGATCTGGCAC	GCGTGTTGAACGTCTCGAAC	57.3	59.4	147
SOD	TCGGGAGATATGTCCGTC	GACACCACAGGCCAAACG	56.0	58.2	126
CAT	GCTGAGATTGAACAGTTGGC	GGTGAGTATCGGGATAGGAG	57.3	59.4	109
GPx	CAGTTTGGGCATCAGGAGAAC	GCATGAAGTTGGGCTCGAAC	59.8	59.4	94

GAPDH: glyceraldehyde-3-phosphate dehydrogenase; ACTB: actin beta; SOD: superoxide dismutase; CAT: catalase; GPx: glutathione peroxidase.

**Table 3 animals-13-00850-t003:** Total phenol contents and antioxidant activities of experimental diets, as measured using different methods (values are given as a mean ± SD of three independent measurements).

	CON	BC-2.5	BC-5	*p*-Value
TPC (mg GAE/g)	3.85 ± 0.15	4.09 ± 0.27	3.84 ± 0.08	ns
DPPH (μmol TE/g)	11.13 ± 0.20	11.14 ± 0.19	11.16 ± 0.17	ns
ABTS (μmol TE/g)	36.42 ± 1.84	34.88 ± 0.23	36.12 ± 0.42	ns
ORAC (μmol TE/g)	113.00 ± 3.8	136.3 ± 4.5	150.70 ± 5.8	<0.001

CON: group without supplemented diet; BC-2.5: group supplemented with 2.5% of bovine colostrum; BC-5: group supplemented with 5.0% of bovine colostrum; GAE: gallic acid equivalent; TE: Trolox equivalent; TPC: total phenol content; DPPH: 2,2-diphenyl-1-picrylhydrazyl; ABTS: 2,20-azinobis(3-ethylbenzthiazoline-6-sulfonic acid); ORAC: oxygen radical absorbance capacity; SD: standard deviation; ns: not significant (*p* > 0.05).

**Table 4 animals-13-00850-t004:** Effect of increasing levels of bovine colostrum supplementation on plasma CAT, GPx and SOD enzymes (values are given as means ± SD).

	CON	BC-2.5	BC-5	*p*-Value
Plasma CAT (U/mL)	11.68 ± 1.86	9.82 ± 1.75	7.66 ± 2.11	0.347
Plasma GPx (U/mL)	196.21 ± 16.67	253.46 ± 33.39	237.27 ± 21.85	0.265
Plasma SOD (U/L)	91.72 ± 8.69	87.38 ± 7.83	77.82 ± 6.85	0.449

CON: group without supplemented diet; BC-2.5: group supplemented with 2.5% of bovine colostrum; BC-5: group supplemented with 5.0% of bovine colostrum; CAT: catalase; GPx: glutathione peroxidase; SOD: superoxide dismutase; SD: standard deviation.

**Table 5 animals-13-00850-t005:** mRNA relative quantification of the antioxidant enzymes CAT, GPx and SOD in the liver and LD muscle of rabbits (values are given as means ± SD).

	Liver			LD muscle			*p*-Value	
	CON	BC-2.5	BC-5	CON	BC-2.5	BC-5	Diet	Tissue
CAT	2.067 ± 0.79	1.995 ± 0.56	1.786 ± 0.35	1.568 ± 1.08	2.659 ± 1.28	2.459 ± 1.18	0.237	0.257
GPx	1.439 ± 0.72	1.244 ± 0.61	0.997 ± 0.24	0.739 ± 0.31	0.806 ± 0.35	0.886 ± 0.31	0.606	0.001
SOD	1.384 ± 1.25	0.722 ± 0.15	0.657 ± 0.15	1.086 ± 0.40	1.315 ± 0.44	1.449 ± 0.37	0.471	0.022

CON: group without supplement diet; BC-2.5: group supplemented with 2.5% of bovine colostrum; BC-5: group supplemented with 5.0% of bovine colostrum; CAT: catalase; GPx: glutathione peroxidase; SOD: superoxide dismutase; SD: standard deviation.

## Data Availability

The data presented in the current study are available from the corresponding authors on reasonable request.

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
