# Peer review of "Antioxidant Activity of Different Tissues from Rabbits Fed Dietary Bovine Colostrum Supplementation"

_animals, 2023, doi:10.3390/ani13050850_

Round 1

Reviewer 1 Report

Comments and Suggestions for Authors

In this paper, the authors report total phenol content (TPC) and antioxidant activities found in bovine colostrum and in the three experimental diets data, ORAC content of dietary group agreed to bovine colostrum supplied data, antioxidant enzymes activity in plasma and antioxidant enzymes gene expression in liver and Longissimus dorsi muscle of rabbits fed dietary bovine colostrum supplementation. The objective of the study was to determine the effect of two dietary increasing levels of bovine colostrum in rabbits on antioxidant enzymes activity in plasma and expression of primary antioxidant enzymes in liver and Longissimus dorsi muscle. The authors demonstrated that antioxidant enzymes activity in plasma and antioxidant enzymes gene expression in liver and Longissimus dorsi muscle of rabbits fed dietary bovine colostrum supplementation have not changed. Based on this study’s finding the use of BC at concentration of 2.5% and 5% did not influence the antioxidant status of rabbits, since muscular and hepatic expression of the antioxidant enzymes has not undergone significant changes. No detrimental or positive alteration has been also observed in the concentration of antioxidant enzymes in plasma. But, they found that mRNA levels were significantly higher in Longissimus dorsi than liver for superoxide dismutase and glutathione peroxidase enzymes. The experiment was balanced with a sufficient number of animals per treatment. The methods of analysis are clear and well detailed. 

*Line 31-34: I suggest:

Further studies considering modification of the length of dietary BC supplementation in doses are required to update the current state of knowledge in rabbits and fully understand the potential value of BC for possible application in farming use.

*Table 1: Put the ingredients % of BC-2.5 and BC-5 diets in table.  Example % of wheat bran in BC-5 diet = 30 * 0.95 = 28.5

*Line 88: Were the 39 rabbits used of both sexes? If so, please explain how you allocated them? If no, please indicate the sex. What was the average age of the animals? Were the groups homogeneous?

*Results: After two months of feeding rabbits with bovine colostrum supplemented diets, it is important to add zootechnical data such as feed intake, daily weight gain or weight gain, feed to gain ratio.

*Line 233: in this sentence ''Antioxidant enzyme activity data are listed in Table 5.’’, is it really Table 5?

*For table 3 to 5, add a column for p values.

*Line 244-246: I suggest: 

A significant tissue related effect has been observed in mRNA levels. Indeed, mRNA levels were significantly higher in LD than liver for SOD (p = 0.022) and GPx (p = 0.001) enzymes (Figure 1).

Author Response

Response to Reviewer 1 Comments

In this paper, the authors report total phenol content (TPC) and antioxidant activities found in bovine colostrum and in the three experimental diets data, ORAC content of dietary group agreed to bovine colostrum supplied data, antioxidant enzymes activity in plasma and antioxidant enzymes gene expression in liver and Longissimus dorsi muscle of rabbits fed dietary bovine colostrum supplementation. The objective of the study was to determine the effect of two dietary increasing levels of bovine colostrum in rabbits on antioxidant enzymes activity in plasma and expression of primary antioxidant enzymes in liver and Longissimus dorsi muscle. The authors demonstrated that antioxidant enzymes activity in plasma and antioxidant enzymes gene expression in liver and Longissimus dorsi muscle of rabbits fed dietary bovine colostrum supplementation have not changed. Based on this study’s finding the use of BC at concentration of 2.5% and 5% did not influence the antioxidant status of rabbits, since muscular and hepatic expression of the antioxidant enzymes has not undergone significant changes. No detrimental or positive alteration has been also observed in the concentration of antioxidant enzymes in plasma. But, they found that mRNA levels were significantly higher in Longissimus dorsi than liver for superoxide dismutase and glutathione peroxidase enzymes. The experiment was balanced with a sufficient number of animals per treatment. The methods of analysis are clear and well detailed. 

We are grateful for your valuable comments and suggestions, which were very useful to improve our paper. We have carefully revised the manuscript and highlighted all changes. Please find our response on the questions asked below.

*Line 31-34: I suggest:

Further studies considering modification of the length of dietary BC supplementation in doses are required to update the current state of knowledge in rabbits and fully understand the potential value of BC for possible application in farming use.

AU: Done as suggested.

*Table 1: Put the ingredients % of BC-2.5 and BC-5 diets in table.  Example % of wheat bran in BC-5 diet = 30 * 0.95 = 28.5

AU: We corrected Table 1.

*Line 88: Were the 39 rabbits used of both sexes? If so, please explain how you allocated them? If no, please indicate the sex. What was the average age of the animals? Were the groups homogeneous?

AU: We added information about age and sex of animals in the Materials & Methods section (Line 90).

*Results: After two months of feeding rabbits with bovine colostrum supplemented diets, it is important to add zootechnical data such as feed intake, daily weight gain or weight gain, feed to gain ratio.

AU: The paper is not focused on performance data, but the primary target of the study was to investigate whether dietary supplementation of bovine colostrum could have an effect on antioxidant enzymes activity of rabbits.

We added information on the initial and final body weight of rabbits in Materials & Methods section (Lines 91 and 100).

*Line 233: in this sentence ''Antioxidant enzyme activity data are listed in Table 5.’’, is it really Table 5?

AU: We corrected with “Table 4”. Thanks for the attention.

*For table 3 to 5, add a column for p values.

AU: Done as suggested. We added column with p-Value in Tables 3, 4 and 5.

We moved the results of the analysis on bovine colostrum from table 3 to the main text (Lines 233-235).

*Line 244-246: I suggest: 

A significant tissue related effect has been observed in mRNA levels. Indeed, mRNA levels were significantly higher in LD than liver for SOD (p = 0.022) and GPx (p = 0.001) enzymes (Figure 1).

AU: Done as suggested (Lines 257-259).

Reviewer 2 Report

Chapter 2.1 is not well described and many data are needed to meet requirements of the reader:

1. Age and weight of the rabbits; mash feed or granules?

2. Source of the colostrum, how it was stored till the trial?

3. How BC was supplemented - liquid or solid, technology used, temperature?

4. From where the tissue samples were taken. Which lobe of liver? Each lobe can have different antioxidant/enzyme activity. Periferal part of lobes may have different activity than inner part. How a place of the collection of LD sample was standartized?

In Table 4, there is a clear dose-dependent tendency to lower plasma CAT and SOD activity. In the case of CAT there is 35% and in SOD there is over 15% decrease in activity and it is not discussed. The CVs in SOD activity are below 10%, so it has to be discussed.

Author Response

Response to Reviewer 2 Comments

Thanks for your effort and time spent on our manuscript. We have made point to point revision according to your constructive suggestions.

Chapter 2.1 is not well described and many data are needed to meet requirements of the reader:

  1. Age and weight of the rabbits; mash feed or granules?

AU: We added requested information (Lines 90-92).

  1. Source of the colostrum, how it was stored till the trial?

AU: The colostrum was obtained from multiparous Holstein-Friesian cows during the first milking, and immediately stored at -20°C until the start of the trial. We clarified this point in the Materials and Methods section (Lines 95-97).

  1. How BC was supplemented - liquid or solid, technology used, temperature?

Bovine colostrum has been added in liquid form before pelleting at 55°C for few seconds, followed by cooling.

We added some details in Materials & Methods section (Line 93).

  1. From where the tissue samples were taken. Which lobe of liver? Each lobe can have different antioxidant/enzyme activity. Periferal part of lobes may have different activity than inner part. How a place of the collection of LD sample was standartized?

AU: We added information as suggested (Lines 104-105).

In Table 4, there is a clear dose-dependent tendency to lower plasma CAT and SOD activity. In the case of CAT there is 35% and in SOD there is over 15% decrease in activity and it is not discussed. The CVs in SOD activity are below 10%, so it has to be discussed.

AU: We considered as trend a threshold value between 0.1 and 0.05 as conventionally accepted.

We added p-values in Table 4 as also requested by Reviewer 1, showing values higher than 0.05, therefore not statistically significant.

Reviewer 3 Report

Review of the paper entitled Antioxidant Activity of Different Tissues from Rabbits Fed Dietary Bovine Colostrum Supplementation by Serra V. et al.

 I suggest the title change reflect the scope of the study (In this paper antioxidant activity in plasma and the gene expression in the liver and muscle was investigated, not the antioxidant activity of different tissues).

There is an antioxidant activity or status not the plasma concentration of antioxidant enzymes (line 17 for example)

There are no research hypotheses.

I have some methodological comments regarding the Materials and Methods section

Line 88: The age of the rabbits should be clarified and I suggest giving the initial and slaughter weight as well as production indicators. Has the clinical condition of rabbits been assessed?

There is no information about bovine colostrum: in what form it was given to rabbits, whether it was an addition of dried colostrum (probably yes), whether it was subjected to heat treatment, how it was standardized, because cattle colostrum varies greatly in composition. It is not clear (description of feeding and table 1) whether the supplement of bovine colostrum was calculated on the basis of dry matter.

Author Response

Response to Reviewer 3 Comments

Review of the paper entitled Antioxidant Activity of Different Tissues from Rabbits Fed Dietary Bovine Colostrum Supplementation by Serra V. et al. 

Many thanks for your time spent to improve the quality of our manuscript.

We have followed your recommendations and answered each of your questions.

I suggest the title change reflect the scope of the study (In this paper antioxidant activity in plasma and the gene expression in the liver and muscle was investigated, not the antioxidant activity of different tissues). There is an antioxidant activity or status not the plasma concentration of antioxidant enzymes (line 17 for example)

AU: We used the term “antioxidant activity” with the intention of including both the gene expression of antioxidant enzymes in liver and muscle, and the antioxidant activity in plasma measured with commercial kits. We supposed to be correct to consider the term “tissues” to include liver, muscle, and blood.

However, if the Reviewer believes it is critical point, we suggest to change the title in: “Antioxidant Activity and Antioxidant Gene Expression in Rabbits Fed Dietary Bovine Colostrum Supplementation”, not yet modified in the revised version of the manuscript.

There are no research hypotheses.

AU: We added the research hypothesis in the Introduction section (Lines 76-78).

-I have some methodological comments regarding the Materials and Methods section:

Line 88: The age of the rabbits should be clarified and I suggest giving the initial and slaughter weight as well as production indicators. Has the clinical condition of rabbits been assessed?

AU: We added information in the Materials and Methods section as suggested (Lines 90-91 and Line 100). Moreover, we added some information about the health status of rabbits in the Discussion section (Lines 377-378).

There is no information about bovine colostrum: in what form it was given to rabbits, whether it was an addition of dried colostrum (probably yes), whether it was subjected to heat treatment, how it was standardized, because cattle colostrum varies greatly in composition. It is not clear (description of feeding and table 1) whether the supplement of bovine colostrum was calculated on the basis of dry matter.

AU: Bovine colostrum has been added in liquid form before pelleting at 55°C for few seconds, followed by cooling.

We added information in Materials & Methods section according also to Reviewer 2 comments (Line 93).

Round 2

Reviewer 2 Report

Presenting of rabbit weights with the precision of hundreds is inadequate and has no sense.

Publication 26 does not provide any data about vitamin C content in colostrum, so the citation:

"The higher antioxidant capacity of colostrum than mature milk depends on presence of higher level of anti-stress vitamins such as Vitamins A (retinol), E (tocopherol), and C present at higher level in colostrum than in normal milk [26]"

is not appropriate.

Author Response

Manuscript animals-2164515

Response to Reviewer 2 comments

1) Presenting of rabbit weights with the precision of hundreds is inadequate and has no sense.

AU: We're not sure we understand what the reviewer means; however, we deleted the decimal after the decimal point from both the initial and final weight, as well as from the standard deviations (Lines 90 and 99), as used in other studies published in Animals (“Abu Afsa et al 2022, The effect of exogenous lysozyme supplementation on growth performance, caecal fermentation and microbiota, and blood constituents in growing rabbits”).

2) Publication 26 does not provide any data about vitamin C content in colostrum, so the citation:

"The higher antioxidant capacity of colostrum than mature milk depends on presence of higher level of anti-stress vitamins such as Vitamins A (retinol), E (tocopherol), and C present at higher level in colostrum than in normal milk [26]"

is not appropriate.

AU: We replaced the previous citation with a more appropriate one (McGrath, B.A.; Fox, P.F.; McSweeney, P.L.; Kelly, A.L. Composition and properties of bovine colostrum: a review. Dairy Sci. Technol. 2016, 96, 133-158, doi:10.1007/s13594-015-0258-x)
